# DeblurDiff: Real-World Image Deblurring with Generative Diffusion Models

**Lingshun Kong**[1,*]**, Jiawei Zhang**[2]**, Dongqing Zou**[3]**, Fu Lee Wang**[4]**,**
**Jimmy S. Ren**[4]**, Xiaohe Wu**[5]**, Jiangxin Dong**[1]**, Jinshan Pan**[1,†]
[1]Nanjing University of Science and Technology [2]SenseTime Research [3]PBVR
[4]Hong Kong Metropolitan University [5]Harbin Institute of Technology

## Abstract

Diffusion models have achieved significant progress in image generation and the pre-trained Stable Diffusion (SD) models are helpful for image deblurring by providing clear image priors. However, directly using a blurry image or pre-deblurred one as a conditional control for SD will either hinder accurate structure extraction or make the results overly dependent on the deblurring network. In this work, we propose a Latent Kernel Prediction Network (LKPN) to achieve robust real-world image deblurring. Specifically, we co-train the LKPN in latent space with conditional diffusion. The LKPN learns a spatially variant kernel to guide the restoration of sharp images in the latent space. By applying element-wise adaptive convolution (EAC), the learned kernel is utilized to adaptively process the blurry feature, effectively preserving the information of the blurry input. This process thereby more effectively guides the generative process of SD, enhancing both the deblurring efficacy and the quality of detail reconstruction. Moreover, the results at each diffusion step are utilized to iteratively estimate the kernels in LKPN to better restore the sharp latent by EAC in the subsequent step. This iterative refinement enhances the accuracy and robustness of the deblurring process. Extensive experimental results demonstrate that the proposed method outperforms state-of-the-art image deblurring methods on both benchmark and real-world images.

## 1 Introduction

Image deblurring aims to recover a sharp image from a blurry observation. Blurring can be caused by various factors, such as camera shake and high-speed movement of the photographed objects. This task is challenging as only the blurry images are available and the blur might be non-uniform. Traditional deblurring methods [17, 16] have made significant progress by utilizing hand-crafted features and priors. However, these methods often struggle to handle complex blur patterns and may produce unsatisfactory results.

Numerous learning-based approaches [25, 3, 31, 1] have been inclined to employ a variety of convolutional neural network (CNN) architectures. Compared with traditional algorithms, CNNs have demonstrated remarkable success. However, the convolution operation is a spatially invariant local operation, which cannot effectively model the spatially variant characteristics of image content and the global contexts for image deblurring. To address the limitations of CNNs, Transformers [32, 27, 12, 10] have been increasingly applied to image deblurring and have achieved commendable performance.

---

*This project is done during the internship at SenseTime Research.
†Corresponding author

39th Conference on Neural Information Processing Systems (NeurIPS 2025).

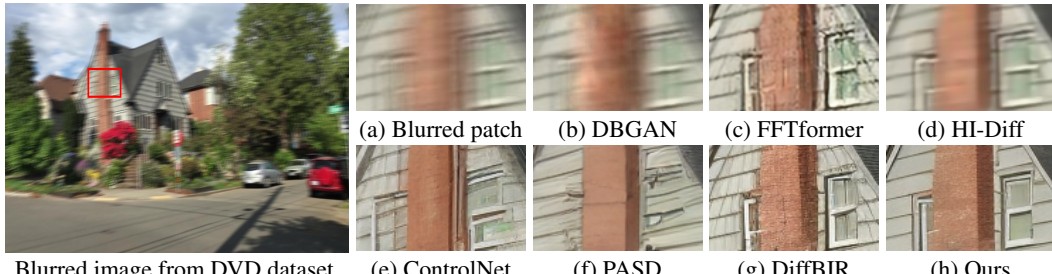

(a) Blurred patch    (b) DBGAN    (c) FFTformer    (d) HI-Diff

Blurred image from DVD dataset    (e) ControlNet    (f) PASD    (g) DiffBIR    (h) Ours

Figure 1: Visual comparison with state-of-the-art image deblurring methods. The results of GAN-based method (b) and diffusion-based method without pretraining (d) still contain significant blur effects. Directly using the blurry image as the conditional input (e) presents significant challenges in effectively extracting structural information. (f) is a method based on pre-trained SD that performs pre-deblurring on the input features, which alters the original information, leading to erratic generation. For (g), it uses the result of the pre-trained FFTformer (c) as the condition. (g) is influenced by the erroneous structures in (c), resulting in generated outputs that retain erroneous artifacts and erroneous structures. In contrast, our approach, guided by the clear structural information provided by LKPN, generates a more distinct and artifact-free image.

Recently, diffusion models have demonstrated outstanding performance in image generation [7, 5]. Some researchers have attempted to utilize Denoising Diffusion Probabilistic Models (DDPMs) for image restoration [18, 30], aiming to leverage their ability to capture complex data distributions and generate detailed images. However, due to the lack of large-scale pre-training, these methods have not demonstrated satisfactory results, particularly when applied to out-of-distribution data, often leading to inferior performance and visually unpleasant artifacts. With the surprising performance of large-scale pre-trained models like Stable Diffusion (SD) in image generation, it has been developed for image restoration [13, 29]. SD models have strong priors on the structure and details of high-quality (HQ) images. However, directly utilizing blurry images as conditional inputs can hinder the extraction of effective structural information, especially in cases of severe blur, ultimately resulting in inaccurately generated structures (Figure 1(e)). While recent methods attempt to mitigate these limitations, their technical trajectories introduce new issues. These methods [13, 29] require training an additional Degradation Removal Model (DRM) to first restore the clear images, and then enhance the details using a ControlNet [36]. This suggests that the restoration results of the entire method are notably influenced by the results of the degradation removal model. When the DRM (implemented via FFTformer [10]) yields inaccurate results (Figure 1(c)), it may adversely affect the diffusion process, potentially resulting in suboptimal performance (Figure 1(g)). Moreover, due to the poor generalization of existing degradation removal models across different datasets, these methods also tend to perform poorly in real-world scenarios.

In this paper, we investigate in-depth the problem of how to leverage pre-trained SD models to assist in real-world image deblurring while reconstructing realistic details and textures. This approach circumvents the direct use of blurry images as conditions, as severe blur can hinder the extraction of accurate structural information, leading to suboptimal final generated structures. Unlike pre-training an additional degradation removal model, which uses restored images as conditions and can be problematic due to poor restoration quality introducing incorrect structures and resulting in erroneous generation, we jointly train a Latent Kernel Prediction Network (LKPN) with the diffusion model. The LKPN, together with Element-wise Adaptive Convolution(EAC), is designed to guide the conditional generation at each step of the diffusion process.

The effectiveness of the LKPN lies in its ability to predict dedicated convolution kernels for each latent pixel, dynamically adjusting the kernel weights based on local content (e.g., edges, textures, and flat regions). These pixel-specific kernels are then applied to the latent blurry image by the EAC, enabling better restoration of clear structures by adaptively addressing distinct blur characteristics at each latent pixel location. This adaptive mechanism not only preserves the necessary information in the input image but also avoids the destruction of structural details, making it particularly advantageous for recovering fine structures and textures. By integrating the LKPN, the diffusion model uses the clear structures from the LKPN as conditional inputs to guide its generation process at each step. This produces more accurate results and enhances both deblurring and detail reconstruction. Furthermore, the intermediate results obtained at each step of the diffusion process are utilized to refine the output of the LKPN, progressively improving the accuracy of the deblurred results by EAC. This iterative refinement generates increasingly clearer conditional inputs, which in turn guide the generation

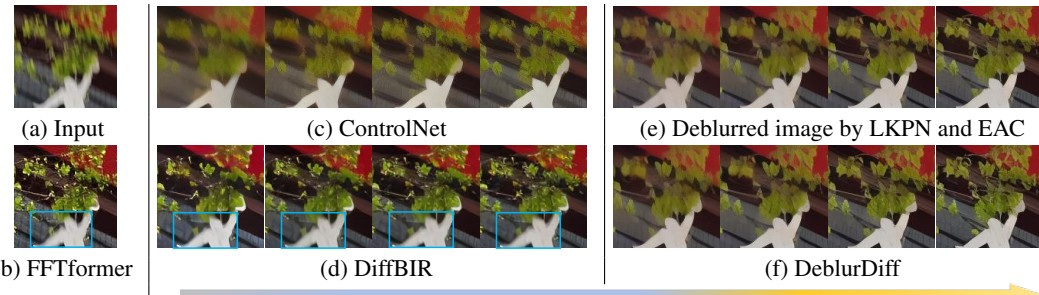

| (a) Input | (c) ControlNet | (e) Deblurred image by LKPN and EAC |
| (b) FFTformer | (d) DiffBIR | (f) DeblurDiff |

Figure 2: Iterative results of the diffusion model. The arrow represents the iterative diffusion process. To visualize this process, we decode the results of each step of the diffusion model through the VAE decoder to the image space. Using the blurry image directly as a conditional input (c) makes the diffusion model struggle to recover clear structures and fine details in (a). For (d), it uses the result of the pre-trained FFTformer (b) as the condition. However, when the results in (b) fail to remove the blur, the generated results will also contain blur (as indicated by the blue box). In contrast, the proposed LKPN can preserve the input information and restore the structure (e) by EAC, thereby guiding the diffusion model to generate better results (f), where the leaves are generated more clearly.

process of the Stable Diffusion (SD) model to achieve iterative improvements. Additionally, the LKPN continuously benefits from the strong prior information provided by SD, enabling it to estimate better kernels to remove blur in the latent space.

The main contributions are summarized as follows: First, we design an architecture based on pre-trained SD models to achieve effective deblurring while reconstructing realistic details and textures. Second, we propose an LKPN architecture that estimates a spatially variant kernel, which is then utilized by the EAC to progressively generate clear structures and preserve the necessary information in the input image throughout the diffusion process. This guides SD to produce more accurate details and structures, thereby enhancing the overall deblurring performance. Finally, we quantitatively and qualitatively evaluate the proposed method on benchmark datasets and real-world images and show that our method outperforms state-of-the-art methods.

## 2 Related Work

**Image deblurring.** Due to the fact that image deblurring is an ill-posed problem, traditional methods [11, 17, 16] often develop various effective priors to constrain the solution space. These manually designed priors can help remove blur. However, they do not fully utilize the characteristics of clear image data, which leads to a struggle in handling complex blur patterns and may produce unsatisfactory results

With the development of deep learning, many learning-based methods have tended to use various CNN architectures for image deblurring. SRN [25] proposes a multi-scale structure that performs image deblurring from coarse to fine. MIMOUnet [3] redesigns the coarse-to-fine structure, significantly reducing the computational cost. NAFNet [1] analyzes the baseline module and simplifies it by removing the activation function, which better facilitates image restoration.

Due to the excellent performance of Transformers in global context exploration and their great potential in many visual tasks, some methods have applied it to image deblurring. Restormer [32] simplifies the baseline module by estimating self-attention in the channel dimension, reducing the computational cost of self-attention Uformer [27] proposes a general U-shaped Transformer model, computing self-attention within local windows to address the image deblurring. FFTformer [10] proposes a frequency-domain based Transformer model and achieves state-of-the-art results.

Although these methods have achieved good deblurring effects, these regression-based methods tend to predict smooth results, with limited ability to depict details.

**Diffusion model.** Denoising Diffusion Probabilistic Models (DDPM) [7] have shown remarkable capabilities in generating high-quality natural images. Some methods [30, 18, 2] have attempted to directly train a diffusion model for image restoration. Rombach et al. [21] extended the DDPM structure to the latent space and conducted large-scale pre-training, demonstrating impressive generative

capabilities. Recently, some researchers have utilized powerful pre-trained generative models, such as SD [21], to address image restoration problems. DiffBIR [13] proposes a two-stage approach, first restoring the degraded image and then using SD to generate details. PASD [29] restores clear images through a Degradation Removal module to provide clear conditional inputs for SD. However, these methods require training an additional image restoration model and then enhance the details through SD. This means that the final results of the SD largely depend on the outcomes of the restoration model. When the degradation removal model produces erroneous results, it may lead to poor performance in the diffusion process.

## 3 Method

Our goal is to leverage the powerful priors of pre-trained Stable Diffusion models to achieve robust image deblurring while simultaneously reconstructing realistic details and textures. Our method iteratively provides clearer guidance to the diffusion model during the diffusion process, thereby progressively restoring clear structures and enhancing details. Figure 3 shows the overview of the proposed method.

### 3.1 Motivation and Preliminaries

**Motivation.** Directly using the blurry image as the condition for SD often results in suboptimal deblurring, as it lacks sufficient structural information to effectively guide the generation process. This makes it difficult for diffusion models to recover clear structures and details, as shown in Figure 2(c). Therefore, providing SD with clear guidance images is necessary. However, directly predicting clear structures from the blurry image is not an easy task and often leads to inaccurate results (Figure 2(b)), which can lead to erroneous structures in the final output of the diffusion model (Figure 2(d)). However, directly predicting a sharp structure from a blurry image is a highly ill-posed problem and often results in inaccurate estimations (Figure 2(b)), which can further lead to erroneous structures in the final output of the diffusion model (Figure 2(d)).

Our work is inspired by existing methods [24, 22, 16], which iteratively estimate the blur kernel and restore the blurred image. Since directly estimating an accurate blur kernel from the blurred image often leads to suboptimal results with residual blur, they first utilize the blurred image along with various priors (e.g., patch priors [24] and dark channel priors [16]) to first estimate an intermediate deblurring result with clear structures. Based on this intermediate result, they then estimate the corresponding blur kernel, which is subsequently used to refine the intermediate deblurring result. This iterative process alternates between kernel estimation and refinement of the intermediate deblurred result, progressively improving the overall deblurring performance. Building upon these ideas, we propose a method that integrates deblur kernel estimation with the clear priors inherent in SD. Specifically, our method uses the priors provided by SD and the blurry input image to first estimate a deblur kernel in the latent space. This kernel is then applied to the blurry image in the latent space to produce a clearer intermediate result, which guides the diffusion model for conditional generation (Figure 2(e)). During the diffusion process, the results generated by the diffusion model are combined with the original blurry image to iteratively estimate a more accurate deblur kernel and progressively refine a clearer guidance image. As the iterations proceed, the enhanced guidance enables SD to progressively recover finer details while preserving the structural integrity of the input. As a result, the diffusion model can generate outputs that are not only sharper and more detailed but also better preserve the original content (Figure 2(f)).

**Denoising Diffusion Probabilistic Models (DDPM)** learn data distributions by progressively denoising a normally distributed variable. The training process involves a forward noise-adding process and a reverse denoising process. At the $t$-th step of the forward diffusion process, a noisy image $x_t$ is generated from the clear image $x_0$ by:

$$x_t = \sqrt{\bar{\alpha}_t}x_0 + \sqrt{1-\bar{\alpha}_t}\epsilon, , \tag{1}$$

where $\epsilon$ denotes the noise sampled from the standard normal distribution $\mathcal{N}(0, I)$, and $\bar{\alpha}_t$ controls the amount of noise added at each step $t$, which is typically a predefined value. In the reverse denoising process, a denoising neural network $\epsilon_\theta(x_t, t)$ is used to predict the noise in $x_t$. After training, the denoising network can iteratively run for $T$ steps starting from noise sampled from a standard normal distribution to generate a clear image. To enhance efficiency, the diffusion process is often moved to

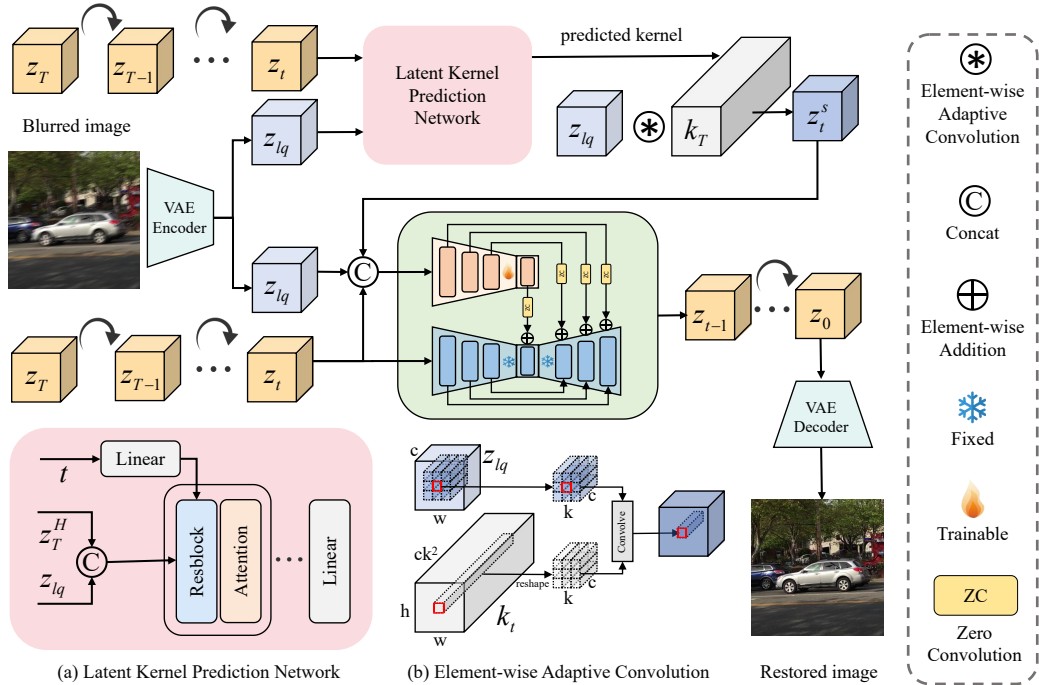

(a) Latent Kernel Prediction Network   (b) Element-wise Adaptive Convolution   Restored image

Figure 3: Overall architecture of the proposed DeblurDiff. It integrates a Latent Kernel Prediction Network (LKPN) with a generative diffusion model to address the challenges of real-world image deblurring. The LKPN leverages the priors from SD and the blurred image to estimate pixel-specific deblurring kernels during the diffusion process. These kernels are applied via Element-wise Adaptive Convolution (EAC) to progressively recover clear structures from blurred images. The refined clear $z^s$ is used as a condition to guide the diffusion process, enabling the model to effectively preserve the input information and structural integrity.

a latent space via a pre-trained Variational Autoencoder (VAE). This latent space is more suitable for likelihood-based generative models as it focuses on the important and semantic parts of the data, operating in a lower-dimensional and computationally efficient space [21]. Specifically, a pre-trained VAE is utilized to transform the clear image $x_0$ into a latent representation $z_0$ by $z_0 = \mathcal{E}(x_0)$, and the forward noising and reverse denoising processes of DDPM are performed in this latent space. The corresponding optimization objective can be simplified to minimizing the denoising loss:

$$\mathcal{L}_{\text{denoise}} = \mathbb{E}_{z_t, \epsilon \sim \mathcal{N}(0,I), t} \left[ \left\| \epsilon - \epsilon_\theta(\mathbf{z}_t, t) \right\|_2^2 \right]. \tag{2}$$

where $z_t$ is the noisy latent representation at step $t$.

## 3.2 LKPN for clear structure guidance.

In the context of blind deblurring, directly using blurry images as conditions can impede the extraction of effective structural information, especially when dealing with significant blur, which can lead to inaccurate final generated structures. To address this challenge, we propose a method that incorporates an LKPN trained simultaneously with the diffusion model. The LKPN dynamically estimates pixel-specific deblurring kernels at each step of the diffusion pipeline, by jointly leveraging the blurred input and intermediate image priors generated during the diffusion process. These estimated kernels are fed into the element-wise adaptive convolution (EAC), which applies content-aware convolutions based on local image structures, allowing the model to effectively address spatially varying blur patterns in the latent space.

Inspired by STFAN [38] and DWNet [6], which predict pixel-wise kernels from single or adjacent frames, we adopt a similar philosophy of spatially adaptive convolution. However, while these methods perform a single, static estimation in the input space, we adapt this idea to the latent space of a pre-trained diffusion model. Our LKPN re-estimates the blur kernel at every denoising step by conditioning on the current, progressively cleaner latent produced by the diffusion backbone, and

Table 1: Quantitative evaluations of the proposed method against state-of-the-art ones on both synthetic and real-world benchmarks. The models marked with an asterisk * indicate that we retrain them on our own training set. The best and second performances are marked in red and blue, respectively. For the RWBI and Real Blurry Images datasets, which lack ground truth (GT) data, we evaluate the performance using only no-reference metrics. We provide comparison results with more methods in the supplementary material.

| Dataset | Metrics | FFTformer* | DBGAN | ResShift | HI-Diff | ControlNet* | PASD* | DiffBIR* | Ours |
|---|---|---|---|---|---|---|---|---|---|
| GoPro | PSNR ↑ | 26.86 | 31.18 | 29.03 | 33.33 | 22.31 | 22.82 | 23.86 | 24.32 |
| | SSIM ↑ | 0.8357 | 0.9182 | 0.8781 | 0.9462 | 0.6547 | 0.6559 | 0.7173 | 0.7375 |
| | LPIPS ↓ | 0.1538 | 0.1120 | 0.0780 | 0.0820 | 0.3292 | 0.2984 | 0.2772 | 0.2191 |
| | FID ↓ | 13.8896 | 10.7629 | 8.8820 | 8.1553 | 37.2749 | 39.0057 | 27.0576 | 17.6948 |
| | NIQE ↓ | 4.1200 | 5.1988 | 4.8367 | 4.6119 | 3.5305 | 2.6567 | 3.4193 | 3.1769 |
| | MUSIQ ↑ | 52.2993 | 42.0985 | 44.2820 | 47.7791 | 59.4246 | 61.5345 | 56.3249 | 61.6369 |
| | MANIQA ↑ | 0.5454 | 0.4976 | 0.5419 | 0.5119 | 0.5746 | 0.5904 | 0.5464 | 0.6134 |
| | CLIP-IQA ↑ | 0.4360 | 0.3788 | 0.4229 | 0.4841 | 0.5869 | 0.5758 | 0.5260 | 0.5966 |
| DVD | PSNR ↑ | 27.07 | 27.78 | 27.78 | 30.31 | 22.03 | 22.23 | 23.49 | 23.74 |
| | SSIM ↑ | 0.8534 | 0.8356 | 0.8420 | 0.8972 | 0.6409 | 0.6440 | 0.7114 | 0.7055 |
| | LPIPS ↓ | 0.1628 | 0.2126 | 0.1249 | 0.1363 | 0.3330 | 0.2968 | 0.2795 | 0.2501 |
| | FID ↓ | 6.7968 | 14.9420 | 7.1787 | 5.7370 | 24.1213 | 22.3637 | 20.2467 | 12.5545 |
| | NIQE ↓ | 3.8562 | 4.8188 | 4.6422 | 5.1858 | 3.4037 | 3.2504 | 3.1357 | 2.7822 |
| | MUSIQ ↑ | 60.1091 | 40.4781 | 52.7551 | 45.5395 | 65.3657 | 68.3299 | 61.6415 | 67.2447 |
| | MANIQA ↑ | 0.6257 | 0.5548 | 0.5744 | 0.5360 | 0.6155 | 0.6409 | 0.5773 | 0.6480 |
| | CLIP-IQA ↑ | 0.5271 | 0.4346 | 0.4831 | 0.4065 | 0.6606 | 0.6528 | 0.5829 | 0.6686 |
| Realblur | PSNR ↑ | 26.94 | 23.91 | 26.30 | 30.18 | 23.77 | 25.02 | 25.50 | 25.71 |
| | SSIM ↑ | 0.8580 | 0.7434 | 0.8140 | 0.9049 | 0.6787 | 0.7642 | 0.7724 | 0.7705 |
| | LPIPS ↓ | 0.1411 | 0.2945 | 0.1249 | 0.0868 | 0.2565 | 0.2075 | 0.1951 | 0.1693 |
| | FID ↓ | 19.8415 | 99.5935 | 21.6440 | 11.4180 | 40.4691 | 38.7507 | 30.4657 | 22.7713 |
| | NIQE ↓ | 4.3473 | 5.3228 | 5.2628 | 5.1437 | 4.6525 | 3.9192 | 4.3053 | 4.2666 |
| | MUSIQ ↑ | 61.5808 | 38.4866 | 49.3209 | 57.1640 | 66.5174 | 61.1498 | 58.7450 | 65.0557 |
| | MANIQA ↑ | 0.6374 | 0.43222 | 0.5373 | 0.6218 | 0.6452 | 0.5994 | 0.5869 | 0.6538 |
| | CLIP-IQA ↑ | 0.5336 | 0.3469 | 0.4521 | 0.5101 | 0.6041 | 0.5457 | 0.5261 | 0.6087 |
| RWBI | NIQE ↓ | 4.4631 | 5.2905 | 5.4446 | 5.3373 | 5.0331 | 4.1973 | 4.2742 | 4.5171 |
| | MUSIQ ↑ | 59.6223 | 42.7631 | 51.0359 | 47.1820 | 62.5079 | 62.1680 | 61.8865 | 66.7505 |
| | MANIQA ↑ | 0.5425 | 0.4852 | 0.4953 | 0.5082 | 0.5758 | 0.5645 | 0.5618 | 0.6260 |
| | CLIP-IQA ↑ | 0.5413 | 0.3645 | 0.5032 | 0.3907 | 0.6199 | 0.5820 | 0.6042 | 0.6849 |
| Real Images | NIQE ↓ | 3.8520 | 4.9338 | 5.4704 | 4.7018 | 4.0978 | 4.4460 | 3.7964 | 3.6628 |
| | MUSIQ ↑ | 52.9290 | 32.0568 | 48.8154 | 43.8702 | 51.5191 | 61.6320 | 53.6088 | 52.9263 |
| | MANIQA ↑ | 0.5170 | 0.4488 | 0.5345 | 0.5722 | 0.5544 | 0.5937 | 0.5564 | 0.5963 |
| | CLIP-IQA ↑ | 0.5026 | 0.3501 | 0.4767 | 0.4545 | 0.5254 | 0.5919 | 0.5384 | 0.5496 |

immediately applies it via EAC. This forms a closed "prior–estimate–restore" loop that turns the conventional static kernel prediction into a dynamic, iterative refinement process, requiring no extra frames yet fully exploiting the generative priors of the diffusion model.

Specifically, we first employ a pre-trained and frozen VAE encoder initialized from SD to encode the blurry image and the clear one into the latent space, obtaining their corresponding latent representations $z_{lq}$ and $z_0$. We follow the Eq.(1) to add noise to $z_0$, obtaining $z_t$. Then the LKPN, which is a U-Net architecture, predicts a spatially variant kernel in latent space at step $t$ :

$$k_t = \text{LKPN}(z_t, z_{lq}, t), \tag{3}$$

where $k_t$ is the predicted kernel at time step $t$. Then $k_t$ is used to refine the blurry image in the latent space by :

$$z_t^s = \text{EAC}(\mathbf{z}_{lq}, k_t), \tag{4}$$

where EAC is the element-wise adaptive convolution in Figure 3 (b) and $z_t^s$ is the deblurred latent result. In the supplementary material, we provide a detailed explanation of the LKPN and EAC architecture.

During training, the LKPN is trained simultaneously with the diffusion model. The LKPN follows the framework of DDPM, imposing constraints simultaneously in both the latent space and the pixel space to progressively optimize the kernel estimation. Specifically, our objective is to minimize the following objective:

$$
\begin{aligned}
\mathcal{L}_{\text{LKPN}} &= \mathcal{L}_{latent} + \mathcal{L}_{pixel}, \\
\mathcal{L}_{latent} &= \mathbb{E}_{z_0, z_{lq}, k_t} \left[ \| z_0 - \text{EAC}(\mathbf{z}_{lq}, k_t) \|_2^2 \right], \\
\mathcal{L}_{pixel} &= \mathbb{E}_{z_0, z_{lq}, k_t} \left[ \| \mathcal{D}(z_0) - \mathcal{D}(\text{EAC}(\mathbf{z}_{lq}, k_t)) \|_2^2 \right],
\end{aligned}
\tag{5}
$$

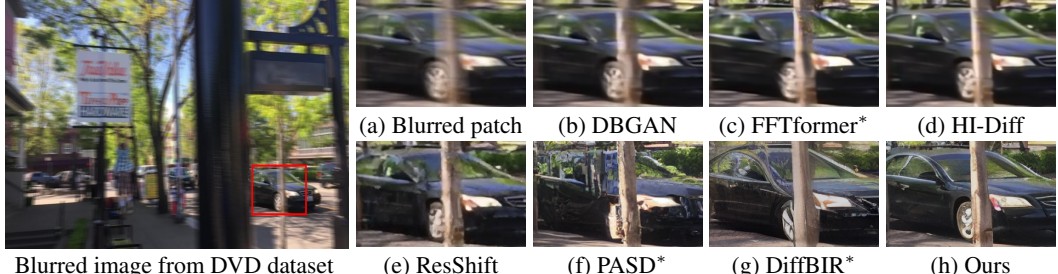

Figure 4: Deblurred results on the DVD dataset [23]. Existing methods struggle to effectively restore clear images. In contrast, our approach not only removes blur but also recovers sharp structures and fine details.

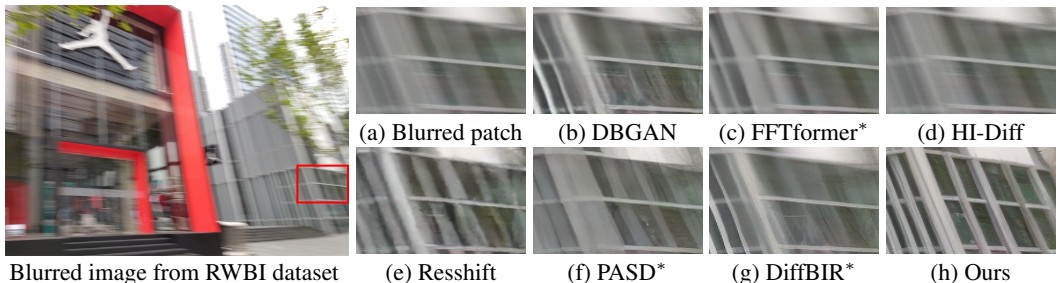

Figure 5: Deblurred results on the RWBI dataset [33]. The structures are not recovered well in (b)-(g). The proposed method generates an image with much clearer structures.

where EAC denotes the element-wise adaptive convolution (EAC) in Figure 3 (b), and $\mathcal{D}$ denotes the pre-trained VAE decoder of SD.

### 3.3 Conditional diffusion for image deblurring.

For the conditional generation network, we follow the training methodology of ControlNet [36], given its demonstrated effectiveness in conditional image generation. Specifically, we adopt the encoder of the UNet in SD as a trainable conditional control branch and initialize the control network by copying weights from the pre-trained SD model. We concatenate $z^s$ recovered by the EAC and $z_{lq}$ as the input to the ControlNet, which is initialized with the weights of SD. During training, the LKPN and ControlNet are jointly trained following the framework of DDPM [7], we minimize the following loss function:

$$\mathcal{L} = \mathcal{L}_{\text{denoise}} \text{ (2)} + \mathcal{L}_{\text{LKPN}} \text{ (5)}. \tag{6}$$

### 3.4 Sampling process of the DeblurDiff.

During the inference stage, the LKPN first estimates the initial spatially variant kernel from random Gaussian noise and the blurred image, which is used to obtain an initial conditional $z_T^s$. The controlled diffusion model uses $z_T^s$ and $z_{lq}$ as conditions to generate an initial clear result $z_{T-1}$. Subsequently, $z_{T-1}$ is fed back into the LKPN to help estimate a more accurate kernel:

$$k_{T-1} = \text{LKPN}(z_{T-1}, z_{lq}, T-1), \tag{7}$$

thereby iteratively optimizing the generated results in subsequent steps. We provide more details in the supplementary material.

This synergy between the LKPN and the diffusion model creates a mutually reinforcing cycle, where clear structural guidance from the LKPN improves the diffusion process, and intermediate results from the diffusion model further refine the deblurred results.

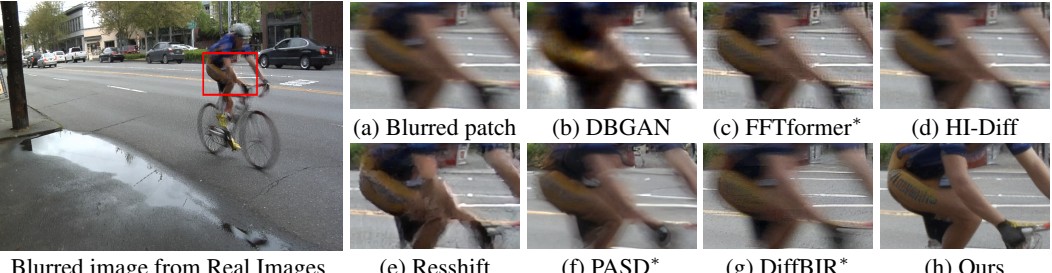

| (a) Blurred patch | (b) DBGAN | (c) FFTformer* | (d) HI-Diff |

Blurred image from Real Images | (e) Resshift | (f) PASD* | (g) DiffBIR* | (h) Ours

Figure 6: Deblurred results on real blurry images dataset [4]. The deblurred results in (b)-(g) still contain significant blur effects. The proposed method generates a clear image.

## 4 Experimental Results

### 4.1 Experimental Settings

**Training Datasets.** Current deblurring datasets are generally small in scale and low in resolution, which is insufficient for effectively training diffusion models. Therefore, we do not use common deblurring datasets such as GoPro [14] as our training set. Instead, we have collected and created a large-scale dataset containing approximately 500,000 data pairs. Our training dataset consists of three parts: (1) Existing deblurring datasets (including MC-Blur [34] and RSBlur [20]). (2) We capture some high-definition video clips, generating blurred and clear data pairs using the same strategy as REDS [15]. (3) We collect a large number of high-definition images and generate various motion blur kernels to synthesize corresponding blurred images. We provide more details about the training dataset in the supplementary material.

**Test Datasets.** We evaluate the proposed DeblurDiff on commonly used image deblurring datasets, including synthetic datasets (GoPro [14], DVD [23]) and Real Blurry Images [4], RealBlur [19], RWBI [33].

**Implementation Details.** We use SD2.1 as the base model. We employ the Adam optimizer [9] to train DeblurDiff with a batch size of 128. The learning rate is set to a fixed value of $5 \times 10^{-5}$. The model is trained for 100,000 iterations using 8 NVIDIA 80G-A100 GPUs.

**Evaluation Metrics.** We employ a range of reference-based and no-reference metrics to provide a comprehensive evaluation of the deblurring performance. For fidelity and perceptual quality assessment, we employ reference-based metrics, including PSNR, SSIM, and LPIPS [37]. For no-reference evaluation, we include NIQE [35], MANIQA [28], MUSIQ [8], and CLIPIQA [26], which assess image quality based on statistical and learning-based approaches. This diverse set of metrics ensures a thorough analysis of both fidelity and perceptual quality.

**Compared Methods.** We compare our DeblurDiff with several state-of-the-art image deblurring methods, which are categorized into two groups: (1) non-diffusion-based methods, including FFT-former [10] and DBGAN [33], and (2) diffusion-based methods, including HiDiff [2], ResShift [30], ControlNet [36], PASD [29], and DiffBIR [13]. We employ the pre-trained FFTformer as the Degradation Removal Module (DRM) in DiffBIR and retrain DiffBIR, as the original DiffBIR framework is not designed for deblurring tasks. We also train a ControlNet and PASD for image deblurring tasks. All the aforementioned retrained methods are trained and tested using the publicly released codes and models of the competing methods, ensuring a fair and consistent comparison under the same dataset and training settings as our proposed method.

### 4.2 Comparisons with the state of the arts

We evaluate our approach on the synthetic and real-world datasets. Table 1 shows the quantitative results. Our method shows strong performance in no-reference metrics, achieving higher scores compared to existing approaches. This indicates that our method excels in perceptual quality and realism, which are critical for real-world applications where ground truth images are often unavailable. Our method achieves lower scores in reference-based metrics compared to HI-DIff. However, HI-Diff does not utilize a pre-trained SD model, thus focusing more on reference-based metrics while lacking generative capabilities, which results in relatively worse performance in no-reference metrics.

Table 2: Effectiveness of each component in the proposed method on the GoPro dataset [14].

| | LKPN | EAC | SD prior for LKPN | PSNR | SSIM | NIQE | MUSIQ | MANIQA | CLIP-IQA |
|---|---|---|---|---|---|---|---|---|---|
| ControlNet | ✗ | ✗ | ✗ | 22.31 | 0.6547 | 3.5305 | 59.4246 | 0.5746 | 0.5869 |
| w/o EAC | ✔ | ✗ | ✔ | 22.57 | 0.6553 | 3.4872 | 58.5845 | 0.5834 | 0.5701 |
| w/o SD for LKPN | ✔ | ✔ | ✗ | 23.14 | 0.6614 | 3.3391 | 59.7763 | 0.5917 | 0.5914 |
| DeblurDiff | ✔ | ✔ | ✔ | **24.32** | **0.7375** | **3.1769** | **61.6369** | **0.6134** | **0.5966** |

Figure 4 shows visual comparisons on the synthetic dataset of DVD [23]. The GAN-based method [33] exhibits significantly inferior deblurring performance, failing to restore clear structures and fine details effectively. Existing diffusion-based methods, such as HiDiff [2] and ResShift [30], fail to achieve satisfactory results in Figure 4(d) and (e) due to their lack of SD priors, which leads to suboptimal generation quality in terms of both structural clarity and detail fidelity. Leveraging the pre-trained FFTformer [10] to preprocess blurred images Figure 4(c) and subsequently applying diffusion-based restoration can partially remove blur. However, since FFTformer sometimes cannot eliminate blur and tends to introduce undesirable artifacts during preprocessing, the final results generated by DiffBIR often exhibit unnatural structures and are inconsistent with the input. In contrast, our method, guided by the LKPN that progressively generates clear structures, produces better results with sharper and more accurate structures.

Figure 5 and Figure 6 present the visual comparison results on the real-world dataset. DBGAN [33] and FFTformer [10] struggle to recover clear structures from severe blur, while Hi-Diff [2] and ResShift [30] fail to reconstruct fine details due to the lack of pre-trained image priors. PASD [29] and DiffBIR [13], which rely on degradation removal models for pre-deblurring, often produce suboptimal results with noticeable artifacts, as these models cannot effectively handle complex blur patterns. In contrast, our method demonstrates better performance in both structural recovery and detail reconstruction. The LKPN progressively generates clear structures and adaptively refines the deblurring process, enabling our approach to achieve high-quality results with minimal artifacts.

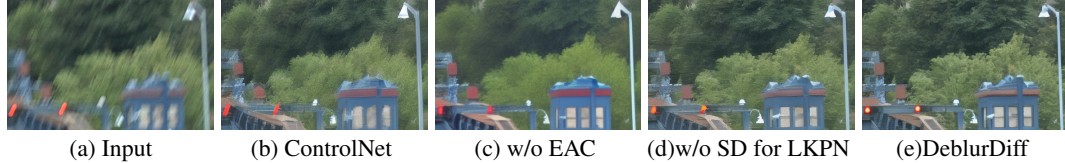

| (a) Input | (b) ControlNet | (c) w/o EAC | (d)w/o SD for LKPN | (e)DeblurDiff |
|---|---|---|---|---|

Figure 7: Effectiveness of the proposed DeblurDiff on image deblurring. ControlNet struggles to recover clear structures due to the lack of explicit structural guidance, resulting in blurred outputs. Without the Stable Diffusion (SD) priors (referred to as w/o SD for LKPN), the LKPN fails to leverage intermediate clear priors for deblurring, leading to artifacts and inconsistencies in the generated results. When the LKPN directly predicts the deblurred result without using EAC (referred to as w/o EAC), the outputs tend to be overly smooth, losing important details and structural information. In contrast, our method effectively recovers sharp structures and fine details while preserving the input information.

## 4.3 Analysis experiments.

The proposed LKPN is used to leverage intermediate clear priors generated during the diffusion process, providing clear structural guidance as conditional inputs to the diffusion model. When the blurry image is directly used as the conditional input (ControlNet for short), the diffusion model struggles to recover clear structures and fine details (Figure 7(b)), particularly in cases of severe blur. This is because the blurry image lacks sufficient structural information to guide the generation process effectively, leading to suboptimal deblurring performance. Table 2 shows the quantitative evaluation results on the real blurry images dataset.

To validate the effectiveness of leveraging clear priors from the diffusion process in the LKPN, we compare our method with the baseline that directly estimates the kernel from the blurry image without utilizing the intermediate results from the diffusion process (w/o SD for LKPN for short). In this baseline, the estimated kernel is applied to the blurry image to generate a deblurred result, which is then used as a conditional input to the diffusion model. Since the LKPN lacks the guidance of intermediate clear priors, its predictions remain static and do not improve over time (Figure 8(a)). As a result, the LKPN struggles to provide accurate structural guidance, leading to structural errors and inconsistencies in the generated results, as shown in Figure 7(d).

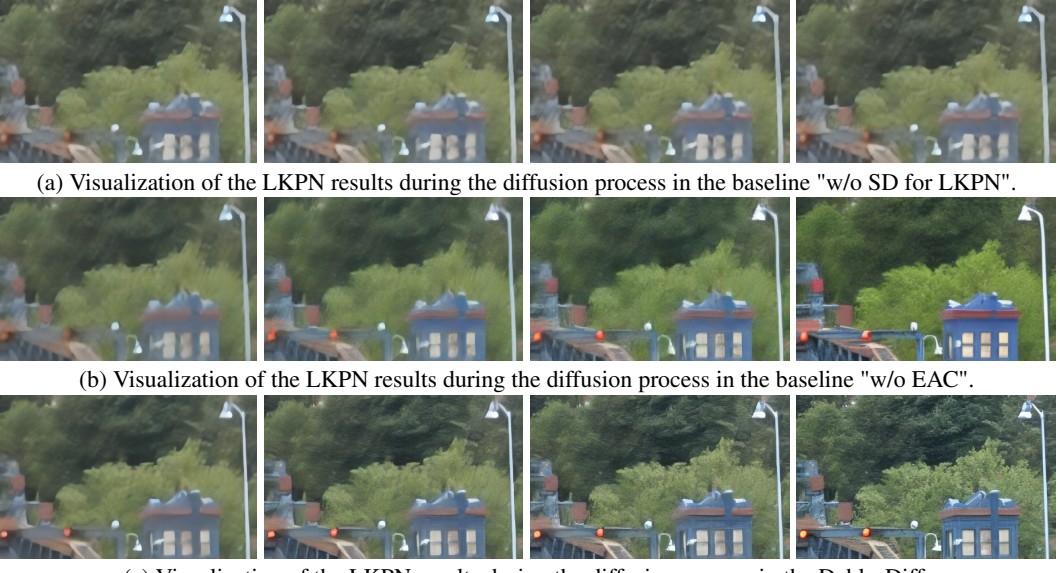

(a) Visualization of the LKPN results during the diffusion process in the baseline "w/o SD for LKPN".

(b) Visualization of the LKPN results during the diffusion process in the baseline "w/o EAC".

(c) Visualization of the LKPN results during the diffusion process in the DeblurDiff.

Figure 8: Iterative results of LKPN. The arrow represents the iterative diffusion process. To visualize this process, we decode the features deblurred by the LKPN to the image space using the VAE decoder in each time step.

We further investigate an alternative design in which the LKPN directly predicts the deblurred latent image—rather than estimating element-wise adaptive kernels—to serve as the conditional input for diffusion. This variant, denoted w/o EAC, performs worse than our full method that predicts kernels and applies EAC to produce clear guidance. This approach fails to adaptively address distinct blur characteristics at each pixel location, resulting in artifacts and inconsistencies in the generated images. Additionally, the generated results of LKPN tend to be overly smooth (Figure 8(b)), and as a consequence, the deblurred images lack sharpness and fine details, leading to suboptimal visual quality and only moderate fidelity. Furthermore, it cannot effectively preserve the input information and structural integrity, resulting in a loss of important details and coherence in the deblurred results (Figure 7(c)).

In contrast, our method leverages intermediate clear priors from the diffusion process to iteratively refine the spatially variant kernels estimated by the LKPN. This iterative refinement enables the LKPN to progressively improve its predictions, generating increasingly accurate deblurring results (Figure 8(c)). The refined kernels are then applied through the Element-wise Adaptive Convolution (EAC), which adaptively addresses distinct blur characteristics at each pixel location, effectively preserving the input information and recovering sharp structures and fine details (Figure 8(e)).

**Limitations.** We develop an effective method that explores the properties of the pre-trained SD for image deblurring. However, we have currently only experimented with the SD2.1 model. In the future, we will extend our method to the SD3.5 model based on DiT and attempt to optimize multi-step diffusion into a single-step diffusion to accelerate model inference.

## 5   Conclusion

In this paper, we propose DeblurDiff, a framework for real-world image deblurring that integrates a Latent Kernel Prediction Network (LKPN) with a generative diffusion model. Our approach addresses the limitations of existing methods by leveraging the priors of pre-trained SD models and introducing an adaptive mechanism to estimate pixel-specific kernels. These kernels are applied through Element-wise Adaptive Convolution (EAC), which adaptively adjusts to local content, enabling the model to preserve input information and structural integrity effectively. Through extensive experiments on both synthetic and real-world benchmarks, we demonstrated that DeblurDiff performs favorably against state-of-the-art methods in terms of structural fidelity and visual quality.

**Acknowledgment.** This work has been supported in part by the National Natural Science Foundation of China (Nos.62272233, U22B2049, 62332010, and 62476067).

**Broader Impact.** This work explores how a diffusion-based deblurring module can be plugged into a pre-trained Stable Diffusion pipeline to restore sharp images on edge devices. The positive implications are immediate: clearer photos on low-cost smartphones, reduced bandwidth in video conferencing, and better visual evidence for journalists or first-responders working in the field. At the same time, the proposed model could be misused, raising concerns about privacy and forensic validity. We therefore advocate responsible disclosure: any public release of the model should be accompanied by (1) visible-watermarking of AI-restored regions, (2) cryptographic provenance metadata, and (3) clear user agreements that prohibit non-consensual enhancement of biometric imagery. Overall, we believe the societal benefit outweighs the risks when these simple safeguards are observed.

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
