# OpenReview forum: "DeblurDiff: Real-Word Image Deblurring with Generative Diffusion Models"
_NeurIPS.cc/2025/Conference — NeurIPS 2025 poster_

### Official Review · Reviewer_TarL · 2025-06-11

**Clarity:** 4
**Significance:** 3
**Originality:** 2
**Rating:** 5
**Confidence:** 5

**Summary:**

This paper propose a novel diffusion-based image deblurring architecture that combining both supervised training and pretrained generative priors, which achieves the SOTA performance on multiple dataset with regard to the visual quality.

**Questions:**

1. Authors mentioned "All the aforementioned retrained methods are trained and tested using the publicly released codes and models of the competing methods". Did you just retrain the DiffBIR or all compared methods. If you retrained all compared methods, then why did the proposed method performs worse on synthetic data. The proposed method also used L2 supervision between ground-truth and output, same as other method.

2. Some related method [https://arxiv.org/pdf/2308.02947] estimate the adaptive kernels on the original image, have you tried this idea instead of on the latent space? And advantage of estimate the kernel in the latent space?

**Ethical Concerns:**

["NO or VERY MINOR ethics concerns only"]

**Final Justification:**

I agree to accept this paper. for the novelty of latent kernel estimation and the solid experiments

**Limitations:**

1. Long inference time makes it inapplicable for real time scenarios
2. The information loss of VAE is inevitable, which may lead to a slight blur on the final result.

**Quality:**

3

**Strengths And Weaknesses:**

Strengths:
1. The method outperforms other SOTA methods on non-reference based metrics.
2. A novel kernel predictor to map the low-quality latent feature to high-quality, avoid the artifacts and blur on the condition.

Weaknesses
1. Lacks the quantitative evaluation of ablation study.
2. In the Figure 13 of the supp, I think "Latent Kernel Prediction Network" should change a name.
3. If all methods are retrained on the proposed dataset, then it does not make sense to me that the proposed method performs worse on PSNR and SSIM, since there is a L2 loss during the training of this method.

---

> ### Author Rebuttal · Authors · 2025-07-30
>
> **1. Lacks the quantitative evaluation of ablation study.**
>
> As real blurred images lack ground-truth references, we employ no-reference metrics for quantitative evaluation. Consequently, Table 2 in the main paper already reports the ablation results with two standard no-reference indices—NIQE and MANIQA. To further strengthen the quantitative evidence, we now provide an extended set of reference-based metrics on the GoPro dataset in the table below.
>
> |                 | PSNR  | SSIM   | LPIPS  | FID     |
> |-----------------|-------|--------|--------|---------|
> | ControlNet      | 22.31 | 0.6547 | 0.3292 | 37.2749 |
> | w/o EAC         | 22.57 | 0.6553 | 0.2914 | 38.7419 |
> | w/o SD for LKPN | 23.14 | 0.6614 | 0.2651 | 25.5312 |
> | DeblurDiff      | **24.32** | **0.7375** | **0.2191** | **17.6948** |
>
> When the EAC module is removed (w/o EAC), the fidelity of DeblurDiff decreases, demonstrating that EAC effectively preserves information from the input image and contributes to improved restoration accuracy. When the features from the pre-trained diffusion model (SD) are not used in LKPN (w/o SD for LKPN), the model lacks the benefit of the iterative refinement guided by the denoised prior, making it difficult for LKPN to provide precise structural guidance. This highlights the importance of the interaction between the degradation estimator and the diffusion model.
>
> **2. In the Figure 13 of the supp, I think "Latent Kernel Prediction Network" should change a name.**
>
> Thank you for this suggestion. We agree that the term “Latent Kernel Prediction Network” in the Figure 13 of the supp is inappropriate. This network should be named “Latent Image Restoration Network”. We will revise the label accordingly.
>
> **3. If all methods are retrained on the proposed dataset, then it does not make sense to me that the proposed method performs worse on PSNR and SSIM, since there is a L2 loss during the training of this method.**
>
> As shown in Table 1 on the GoPro dataset, the PSNR of non-diffusion methods (e.g., FFTformer) is generally higher than that of diffusion-based approaches, with FFTformer outperforming all diffusion-based methods in PSNR. This is because non-diffusion models, such as CNNs or transformers, are typically optimized directly for pixel-level accuracy. In contrast, the training objective of diffusion models focuses more on generating high-quality, perceptually realistic images rather than pixel-level precise matching. Although L2 loss is included during training, the core optimization goal and the stochastic nature of the generative process may result in relatively lower performance on pixel-level metrics like PSNR and SSIM compared to methods specifically designed for these metrics. This is an inherent difference in the design objectives between generative and discriminative models.
>
> **4. Did you just retrain the DiffBIR or all compared methods. If you retrained all compared methods, then why did the proposed method performs worse on synthetic data. The proposed method also used L2 supervision between ground-truth and output, same as other method.**
>
> We provide a comprehensive comparison of all retrained methods in Table 1 of the supplemental material
>
> The lower PSNR/SSIM scores for our diffusion-based method, despite using L2 supervision, stem from fundamental differences:
>
> 1. Optimization Focus: CNNs or Transformers (FFTformer) directly minimize the pixel-wise L2 error between the predicted output image and the ground truth, which inherently optimizes for high PSNR and SSIM. In contrast, our diffusion model is trained to predict the noise added to a latent representation at each denoising step. The L2 loss is applied between the predicted noise and the actual noise, not directly on the final reconstructed image. This indirect optimization prioritizes learning the data distribution and generating perceptually realistic details, often at the expense of perfect pixel-level accuracy. As a result, while our method produces visually superior results, its PSNR and SSIM metrics may be lower than those of CNNs that are explicitly trained to minimize pixel-level reconstruction error.
>
> 2. Model Prior: Diffusion models use a powerful generative prior, which can produce more realistic textures but may sacrifice slight pixel fidelity compared to CNNs or Transformers optimized for these metrics. When compared to the pre-trained diffusion-based approaches (ControlNet, DiffBIR, PASD), our method achieves higher scores in both reference-based (PSNR, SSIM) and no-reference metrics. Specifically, on the GoPro, DVD, and RealBlur datasets, our method consistently outperforms ControlNet, DiffBIR, and PASD in both PSNR and SSIM (0.44dB higher in PSNR on the GoPro dataset), demonstrating better restoration fidelity. This indicates that our method is more effective at preserving the information from the input image.
>
>
> **5. Some related method estimate the adaptive kernels on the original image, have you tried this idea instead of on the latent space? And advantage of estimate the kernel in the latent space?**
>
> Since diffusion models operate in the latent space to significantly reduce computational cost, we design our approach to be compatible with this paradigm. Therefore, we perform kernel estimation directly in the latent space instead of on the original high-resolution image. Estimating the kernel on the original image would not only contradict this efficiency principle but also lead to dramatically higher memory and processing demands, as the resolution is 8 times larger. Additionally, it would require continuous and expensive VAE encoding/decoding during both training and inference.
>
> Estimating the kernel in the latent space is advantageous because it significantly reduces computational overhead while maintaining acceptable accuracy. The lower resolution makes the process more efficient and avoids repeated VAE operations.
>
> We will cite and discuss the distinctions from these papers in the revised manuscript.
>
> **6. Long inference time makes it inapplicable for real time scenarios**
>
> This is indeed a limitation of our method. As discussed in L295–L297 of the main paper, we plan to optimize the multi-step diffusion into a single-step approach to accelerate inference.
>
> **7. The information loss of VAE is inevitable, which may lead to a slight blur on the final result.**
>
> Traditional VAEs often compress an image into a single latent vector, which can lead to severe information loss and blurry reconstructions due to the reliance on pixel-wise L2 loss. In contrast, the VAEs used in pre-trained diffusion models (Stable Diffusion and FLUX) are specifically retrained and designed differently. They do not compress the image into a single vector, but rather into a spatial latent feature map of size h/8 x w/8 (where h and w are the original image dimensions), which preserves much more spatial information.
>
> Furthermore, their training process incorporates advanced loss functions, such as perceptual loss and adversarial loss, in addition to reconstruction loss. This design prioritizes perceptual quality and detail preservation, resulting in significantly clearer and more realistic reconstructions compared to traditional VAEs. While some information loss is still inherent, the VAEs in modern diffusion models are optimized to minimize this issue.
>
> We acknowledge this limitation and believe it's an area where future research on diffusion models can focus.

---

> > ### Comment · Reviewer_TarL · 2025-08-04
> >
> > I think the authors addressed my concerns

---

### Official Review · Reviewer_34aF · 2025-06-29

**Clarity:** 3
**Significance:** 3
**Originality:** 3
**Rating:** 5
**Confidence:** 5

**Summary:**

This paper presents DeblurDiff, a novel real-world image deblurring framework that integrates a Latent Kernel Prediction Network (LKPN) with pre-trained Stable Diffusion (SD) models. The LKPN predicts pixel-wise adaptive kernels to refine blurry latent representations via Element-wise Adaptive Convolution (EAC), providing improved structural guidance for the diffusion process. Extensive experiments demonstrate superior perceptual quality over state-of-the-art CNN-, Transformer-, and diffusion-based deblurring methods.

**Questions:**

Please refer to the weakness part.

**Ethical Concerns:**

["NO or VERY MINOR ethics concerns only"]

**Final Justification:**

Thanks to the author's reply, I think this paper is acceptable and I maintain the score.

**Limitations:**

Yes. The authors mention limitations regarding SD model versions and runtime efficiency.

**Paper Formatting Concerns:**

None.

**Quality:**

3

**Strengths And Weaknesses:**

+ Strong design combining SD priors with an adaptive kernel estimation strategy; ablation studies (Table 2) and multiple benchmarks validate the method.
+ The paper is generally well-written with clear motivation, figures, and architectural illustrations (Fig.3).
+ Shows practical potential by addressing limitations of existing degradation-removal pipelines.

- The improvement in reference-based metrics (PSNR, SSIM) over prior methods is modest, despite substantial complexity.
- Limited discussion on computational overhead of iterative kernel refinement during inference.

---

> ### Author Rebuttal · Authors · 2025-07-30
>
> **1. The improvement in reference-based metrics (PSNR, SSIM) over prior methods is modest, despite substantial complexity.**
>
> Compared to the transformer-based FFTformer, our PSNR is lower, which is partly due to the inherent trade-off in diffusion models between perceptual quality and pixel-level accuracy. Our method prioritizes generating visually realistic and detailed results, especially on real-world images, over maximizing PSNR/SSIM.
>
> When compared to the pre-trained diffusion-based approaches (ControlNet, DiffBIR, PASD), our method achieves higher scores in both reference-based (PSNR, SSIM) and no-reference metrics. Specifically, on the GoPro, DVD, and RealBlur datasets, our method consistently outperforms ControlNet, DiffBIR, and PASD in both PSNR and SSIM (0.44dB higher in PSNR on the GoPro dataset), demonstrating better restoration fidelity. This indicates that our method is more effective at preserving the information from the input image.
>
> Furthermore, on all four benchmarks—GoPro, DVD, RealBlur, and RWBI—our method achieves the best performance in the no-reference metrics MANIQA and CLIP-IQA, indicating better perceptual quality. Our method prioritizes generating visually realistic and detailed results, especially on real-world images, over maximizing PSNR/SSIM, which aligns with the strengths of the diffusion framework.
>
> **2. Limited discussion on computational overhead of iterative kernel refinement during inference.**
>
> We analyzed the computational overhead, including runtime and GPU memory, in L86–L92 and Table 4 of the Supplemental Material.  As shown in the following table, we provide a comprehensive comparison of other methods in terms of parameters, inference time, and FLOPs.
>
> |           | FFTformer | ResShift | DiffBIR | PASD  | Ours  |
> |-----------|-----------|----------|---------|-------|-------|
> | Params(M) | 16.6      | 119      | 1717    | 1900  | 1845  |
> | FLOPs(G)  | 131       | 5491     | 24234   | 29125 | 25526 |
> | Times(s)  | 0.8       | 15       | 55      | 52    | 46    |
>
> Compared to FFTformer, our method is slower in inference time, as FFTformer is not a diffusion-based method and thus inherently faster. ReShift, which does not use a pre-trained diffusion model, also has a runtime advantage. However, both methods achieve inferior visual quality compared to the approaches that does not use a pre-trained diffusion model.
>
> Among all methods that leverage pre-trained diffusion models (ControlNet, DiffBIR, PASD, and ours), our method achieves a favorable trade-off between speed and performance. With an inference time of 46 seconds, our method is slightly slower than ControlNet (40 seconds) but 16.4\% faster than DiffBIR (55 seconds). In terms of GPU memory, our method uses 7.8 GB, which is slightly higher than ControlNet (7.6 GB) but lower than DiffBIR (8.0 GB). Additionally, our method has fewer parameters and lower FLOPs than PASD.
>
> This demonstrates that our iterative kernel refinement does not introduce significant computational overhead. More importantly, among all pre-trained diffusion-based methods, our approach achieves both higher fidelity (as shown by superior PSNR and SSIM in Table 1) and better perceptual quality. This indicates that our method delivers state-of-the-art performance without incurring excessive computational cost.

---

> > ### Comment · Reviewer_34aF · 2025-08-06
> >
> > Thanks to the authors' efforts, my concerns has been solved.

---

### Official Review · Reviewer_Xgdm · 2025-07-02

**Clarity:** 3
**Significance:** 2
**Originality:** 2
**Rating:** 3
**Confidence:** 4

**Summary:**

Directly using a blurry image or pre-deblurred one as a conditional control for Stable Diffusion will either hinder accurate structure extraction or make the results overly dependent on the deblurring network. In this work, author proposes a Latent Kernel Prediction Network (LKPN) to achieve robust real world image deblurring.

**Questions:**

1.The compared methods are still too outdated — DiffBIR was released in 2023. I hope the authors can compare with more recent SOTA methods to truly demonstrate the effectiveness of their approach.

2.The author's method is compared with several others in Table 1. From what I see, these methods were all released in 2023. Even so, I find that the performance of the author's method is still not SOTA. I'm also confused because the author's PSNR, SSIM, and LPIPS scores are not as good as those of Hi-Diff, MUSIQ, NIQE, CLIP-IQA, or even PASD (also from 2023). So, what justifies the claim that your method is advanced?

3.Please provide a comparison of all methods in Table 1 in terms of the number of parameters and inference time.

4.The NIQE and MANIQA metrics used in Table 2 are no-reference quality assessment indicators, and they exhibit large fluctuations. Furthermore, they cannot accurately evaluate visual quality. Metrics like PSNR, SSIM, and LPIPS would be more appropriate here.

5.The novelty of the paper is not very strong. Degradation estimation in Latent Diffusion Models (LDM) has already been explored by many image restoration methods, such as:

[1] DiffIR: Efficient diffusion model for image restoration

[2] DeeDSR: Towards Real-World Image Super-Resolution via Degradation-Aware Stable Diffusion

[3] Degradation-Guided One-Step Image Super-Resolution with Diffusion Priors

**Ethical Concerns:**

["NO or VERY MINOR ethics concerns only"]

**Final Justification:**

For A1, the publication note for TPAMI 2024 indicates that PASD was submitted in 2023, while the submission note for ECCV 2024 states diffbir was released before in February 2024. In fact, even the latest version of DiffBIR doesn't show much difference from the version released in June 2023. These papers have been around for at least a year and a half. In such a rapidly developing field, model performance improves every three months, so the methods compared by the authors are too outdated, and I believe they lack any convincing power.
2.For A2, although the authors provided some explanations, they cannot change the fact that its performance is inconsistent. In my view, this is merely an incremental improvement and enhancement.

3.For A3, it seems that the methods you compared in Table 1 are not limited to the few you’ve listed. Please provide a comparison of parameters and inference time for a few other methods, such as ControlNet, DBGAN, and HI-Diff. This would allow for a better determination of the advantages of your approach.

4.For A5, after hearing your explanation, I still think it is just an incremental improvement, and the performance does not show significant progress compared to those older methods.

I believe the issues mentioned above significantly undermine the credibility of this paper, and it does not meet the acceptance standards. Therefore, I maintain the decision to reject it.

**Limitations:**

yes

**Quality:**

3

**Strengths And Weaknesses:**

The paper is well written.

---

> ### Author Rebuttal · Authors · 2025-07-30
>
> **1. The compared methods are outdated**
>
> Thank you for your suggestion. We would like to clarify that the version of DiffBIR we compare against is not outdated. While the initial arXiv submission was in August 2023, the method has been significantly updated and was accepted to ECCV 2024. We compare with this latest and most competitive version of DiffBIR.
>
> Similarly, PASD was also published at ECCV 2024, and for ReShift, we use the updated results from its TPAMI 2024 publication. Therefore, our comparisons are based on the most recent and up-to-date versions of these methods, ensuring a fair and current evaluation.
>
> Importantly, the final published versions of both DiffBIR and PASD (in ECCV 2024) contain significant updates and improvements compared to their initial arXiv submissions, making them strong and contemporary baselines for comparison.
>
> Furthermore, we have evaluated the "Efficient Visual State Space Model for Image Deblurring (EVSSM)", which was published at CVPR 2025, on the benchmarks. The results are shown in the following table. EVSSM$^*$ indicates that we retrain EVSSM on our training set.
>
> Our method also achieves better performance on no-reference metrics compared to the latest image deblurring method from CVPR 2025 on real-world images, demonstrating better visual quality.
>
> | RWBI dataset | NIQE   | MUSIQ   | MANIQA | CLIPIQA |
> |--------------|---------|---------|--------|---------|
> | EVSSM (CVPR 2025)        | 5.0161 | 46.9781 | 0.4968 | 0.3827  |
> | EVSSM$^*$     | 4.8744 | 57.1694 | 0.5611 | 0.5686  |
> | DeblurDiff   | **4.5171** | **66.7505** | **0.6260** | **0.6849**  |
>
> | Real Images dataset | NIQE   | MUSIQ   | MANIQA | CLIPIQA |
> |--------------|--------|---------|--------|---------|
> | EVSSM (CVPR 2025)        | 4.6822 | 38.5475 | 0.5362 | 0.4141  |
> | EVSSM$^*$    | 3.8132 | 51.9340 | 0.5517 | 0.5176  |
> | DeblurDiff   |  **3.6628** | **52.9263** | **0.5963** | **0.5496**  |
>
> **2. The proposed method is not the best on some metrics.**
>
> Our goal is to develop a diffusion-based framework that leverages the powerful generative prior of pre-trained models to produce high-fidelity and perceptually realistic results for image restoration. While pixel-level metrics like PSNR are important, our primary focus is on achieving a superior balance between fidelity and perceptual quality.
>
> It is important to note that Hi-Diff is fundamentally a regression-based method built on a transformer architecture, which only incorporates features from diffusion models as guidance, rather than using a diffusion process to generate the final image. In contrast, our method is a fully diffusion-based framework.
>
> Among all methods that leverage pre-trained diffusion models (ControlNet, DiffBIR, PASD, and DeblurDiff), our method achieves the highest PSNR, SSIM, and LPIPS scores on synthetic datasets (GoPro, DVD, RealBlur), as shown in Table 1 of the main paper, demonstrating better fidelity. This indicates that our method is more effective at preserving the information from the input image.
>
> For no-reference metrics, Table 1 of the main paper shows that our method achieves the best MANIQA and CLIP-IQA scores on all four benchmarks: GoPro, DVD, RealBlur, and RWBI. On the RealImages dataset, our method also achieves the highest NIQE and MANIQA.
>
> We acknowledge that on the Real Images dataset, our MUSIQ and CLIP-IQA scores are slightly lower than those of PASD. However, our overall state-of-the-art performance across the majority of metrics and datasets demonstrates the effectiveness and advancement of our approach.
>
>
> **3. Please provide a comparison of all methods in Table 1 in terms of the number of parameters and inference time.**
>
> We have provided runtime information for some methods in Table 4 of the Supplemental Material.
>
> We provide a more comprehensive comparison of other methods, including the number of parameters and inference time, in the following table.
>
> |           | FFTformer | ResShift | DiffBIR | PASD  | Ours  |
> |-----------|-----------|----------|---------|-------|-------|
> | Params (M) | 16.6      | 119      | 1717    | 1900  | 1845  |
> | FLOPs (G)  | 131       | 5491     | 24234   | 29125 | 25526 |
> | Times (s)  | 0.8       | 15       | 55      | 52    | 46    |
>
> As discussed in L86–L92 of the Supplemental Material, FFTformer is not a diffusion-based method, which makes it faster but results in inferior visual quality compared to diffusion-based approaches. ReShift does not use a pre-trained diffusion model, giving it some advantages in runtime over methods that do; however, its visual performance is worse than diffusion-based methods such as DiffBIR, PASD, and DeblurDiff. Among all methods that leverage pre-trained diffusion models, our approach achieves both higher fidelity (as shown by better PSNR and SSIM in Table 1) and better perceptual quality. Moreover, our method has fewer parameters and lower FLOPs than PASD, and runs faster than both DiffBIR and PASD.
>
> **4. The NIQE and MANIQA metrics used in Table 2 are no-reference quality assessment indicators, and they exhibit large fluctuations. Furthermore, they cannot accurately evaluate visual quality. Metrics like PSNR, SSIM, and LPIPS would be more appropriate here.**
>
> Since Table 2 in the main paper evaluates results on real-world images where ground truth (GT) is unavailable, metrics like PSNR and SSIM cannot be computed. To provide a comprehensive comparison, we also report PSNR, SSIM, and LPIPS on synthetic datasets (GoPro) where GT is available, as shown in the following table.
>
> |                 | PSNR  | SSIM   | LPIPS  | FID     |
> |-----------------|-------|--------|--------|---------|
> | ControlNet      | 22.31 | 0.6547 | 0.3292 | 37.2749 |
> | w/o EAC         | 22.57 | 0.6553 | 0.2914 | 38.7419 |
> | w/o SD for LKPN | 23.14 | 0.6614 | 0.2651 | 25.5312 |
> | DeblurDiff      | **24.32** | **0.7375** | **0.2191** | **17.6948** |
>
> When the EAC module is removed (w/o EAC), the fidelity of DeblurDiff decreases, demonstrating that EAC effectively preserves information from the input image and contributes to improved restoration accuracy. When the features from the pre-trained diffusion model (SD) are not used in LKPN (w/o SD for LKPN), the model lacks the benefit of the iterative refinement guided by the denoised prior, making it difficult for LKPN to provide precise structural guidance. This highlights the importance of the interaction between the degradation estimator and the diffusion model.
>
> **5. The novelty of the paper is not very strong. Degradation estimation in Latent Diffusion Models (LDM) has already been explored by many image restoration methods**
>
> Methods like DiffIR operate directly in the pixel space, while DeeDSR, S3Diff, and PASD, though performing estimation in the latent space, directly estimate degradation from the corrupted input. This makes their results highly dependent on the accuracy of the initial degradation estimation.
>
> In contrast, our method introduces two key innovations:
>
> 1. Iterative Refinement with a Clean Prior: Methods like PASD, DeeDSR, and S3Diff perform a single, static degradation estimation at the beginning of the process, typically based solely on the corrupted latent input. This makes their entire restoration pipeline critically dependent on the accuracy of this initial, one-off estimate. If the initial estimation is poor, the error propagates irreversibly through the diffusion process, leading to suboptimal results.
> In contrast, our method introduces a dynamic, iterative refinement loop. We do not estimate the degradation once, but re-estimate it at every diffusion step. Crucially, our estimation leverages not just the corrupted input, but also the progressively denoised and cleaner latent features generated by the diffusion model itself. This denoised output serves as a powerful "clean prior," providing rich, high-quality structural information that guides a more accurate degradation estimate for the next step. This closed-loop feedback mechanism allows the degradation estimate to evolve and improve over time, correcting early mistakes and leading to significantly more robust and accurate results.
>
> 2. Deblur Kernel-Guided Reconstruction: Existing methods employ the estimated degradation as a generic conditioning signal (e.g., via concatenation as an auxiliary input). In contrast, our approach leverages the predicted deblurring kernel in a more precise and effective manner by integrating it through an Element-wise Adaptive Convolution (EAC) module, which performs feature-adaptive modulation in latent space. This allows the model to apply spatially adaptive deblurring operations. Such a design ensures tight coupling between the restoration process and the estimated degradation, thereby enhancing reconstruction fidelity and mitigating artifact generation.
>
> These design choices result in a more accurate estimation and higher-quality restoration, representing a meaningful advancement over prior approaches. We will cite and discuss the distinctions from these papers in the revised manuscript.

---

> > ### Author Response · Authors · 2025-08-06
> >
> > Dear Reviewer,
> >
> > As the discussion period is coming to a close, please let us know if you have any additional questions or comments regarding our previous rebuttal. We would be happy to provide further clarification or explanations if needed.
> >
> > Best,
> >
> > Authors

---

> > > ### Comment · Reviewer_Xgdm · 2025-08-06
> > > **Response to Author**
> > >
> > > 1. For A1, the publication note for TPAMI 2024 indicates that PASD was submitted in 2023, while the submission note for ECCV 2024 states diffbir was released before in February 2024. In fact, even the latest version of DiffBIR doesn't show much difference from the version released in June 2023. These papers have been around for at least a year and a half. In such a rapidly developing field, model performance improves every three months, so the methods compared by the authors are too outdated, and I believe they lack any convincing power.
> > >
> > > 2.For A2, although the authors provided some explanations, they cannot change the fact that its performance is inconsistent. In my view, this is merely an incremental improvement and enhancement.
> > >
> > > 3.For A3, it seems that the methods you compared in Table 1 are not limited to the few you’ve listed. Please provide a comparison of parameters and inference time for a few other methods, such as ControlNet, DBGAN, and HI-Diff. This would allow for a better determination of the advantages of your approach.
> > >
> > > 4.For A5, after hearing your explanation, I still think it is just an incremental improvement, and the performance does not show significant progress compared to those older methods.
> > >
> > > I believe the issues mentioned above significantly undermine the credibility of this paper, and it does not meet the acceptance standards. Therefore, I maintain the decision to reject it.

---

> > > > ### Author Response · Authors · 2025-08-06
> > > >
> > > > Thank you for your response.
> > > >
> > > > 1. Regarding the issue of comparison methods, to the best of our knowledge, there are currently no additional methods that utilize pre-trained diffusion models to address the deblurring problem. Therefore, we have retrained the method from ECCV 2024 for deblurring purposes and included it in our comparison. We have added the latest method (EVSSM) from CVPR 2025 in the rebuttal, and our method outperforms it, as demonstrated in the rebuttal. Do you have any suggestions on other methods we should compare with?
> > > >
> > > > 2. In terms of performance, among all the methods utilizing pre-trained diffusion models (ControlNet, DiffBIR, PASD, and DeblurDiff), our method achieves the highest PSNR, SSIM, and LPIPS scores on the synthetic datasets (GoPro, DVD, RealBlur), as shown in Table 1. For no-reference metrics, Table 1 in the main paper shows that our method achieves the best MANIQA and CLIP-IQA scores on four benchmarks (GoPro, DVD, RealBlur, and RWBI). On the RealImages dataset, our method also achieves the highest NIQE and MANIQA. Our method is not, as you said, "the performance does not show significant progress compared to those older methods." We don't understand what you mean by "incremental improvement." Could you please elaborate on that?
> > > >
> > > > 3. Regarding the additional metrics, we have provided the additional metrics you requested in the table below.
> > > >
> > > > |           | DBGAN | FFTFormer | Hi-DIff | ResShift | ControlNet | DiffBIR | PASD  | Ours  |
> > > > |-----------|-------|-----------|---------|----------|------------|---------|-------|-------|
> > > > | Params(M) | 1.8   | 16.6      | 24.0    | 119      | 1699       | 1717    | 1900  | 1845  |
> > > > | FLOPs(G)  | 39    | 131       | 125     | 5491     | 23174      | 24234   | 29125 | 25526 |
> > > > | Times(s)  | 0.5   | 0.8       | 0.9     | 15       | 40         | 55      | 52    | 46    |
> > > >
> > > > We hope to have further discussions with you to address your concerns.

---

> ### Author Response · Authors · 2025-08-08
>
> Dear Reviewer,
>
> Have you had a chance to review our new response? Does it address your concerns? We have already responded to your questions and compared the latest methods (EVSSM in CVPR 2025) in our initial rebuttal. We have also provided explanations regarding the performance of our method. If there are still any questions, we hope you can let us know.
>
>
> Best,
>
> Authors

---

### Official Review · Reviewer_qdjB · 2025-07-11

**Clarity:** 2
**Significance:** 3
**Originality:** 2
**Rating:** 5
**Confidence:** 4

**Summary:**

This paper presents DeblurDiff, a new diffusion based model for real-world image deblurring. The key contribution is alternating diffusion of two entities: latent kernel and latent image feature. The Latent Kernel Prediction Network (LKPN) predicts spatially variant deblurring kernels within the latent space and it is trained in conjunction with a conditional diffusion model. These kernels are then applied to the latent representation of the blurry image using Element-wise Adaptive Convolution (EAC), producing a cleaner conditional input for the diffusion process. This iterative process is used to refine the kernel prediction and latent image prediction for the subsequent step. The model achieves state-of-the-art results, especially in terms of perceptual quality, across both synthetic and real-world deblurring datasets.

**Questions:**

Operations such as attention have become the new norm and they have essentially replaced convolution operations in image restoration literature. Did the project explore any alternatives to convolution operation?

I wonder if a DRM would also benefit the proposed approach. Intuitively, adding a predeblurred image (using a DRM) to the conditioning images could further boost the accuracy of the model. Since the denoiser of ControlNet is capable of accepting multiple latents as input (in concatenated format), maybe we could also supply the latents extracted from a predeblurred image This could further improve scores since exiting methods such as PASD using such predeblurred image are already doing well across benchmarks, specially on real images (Table 1)

The paper mentions that a pre-trained FFTformer is used as the DRM in DiffBIR. Kindly clarify whether or not the FFTformer used in DiffBIR is retrained on the proposed dataset. If not, the comparison with DiffBIR would not be completely fair.

Kindly also address the key weaknesses mentioned above.

**Ethical Concerns:**

["NO or VERY MINOR ethics concerns only"]

**Final Justification:**

Using diffusion based Kernel Prediction in the latent domain is an effective solution to a key problem in diffusion based deblurring methods: conditioning without losing important detail or introducing artifacts. The perceptual quality of results shown is quite impressive. The proposed large scale dataset (if released) is valuable to the future development in the deblurring community. The details and insights validate the approach! The only negative aspect is lack of fundamental novelty in the proposed modules.

**Limitations:**

Yes

**Quality:**

3

**Strengths And Weaknesses:**

Strengths:

Using diffusion based Kernel Prediction in the latent domain to iteratively refine the input for the diffusion based restoration model is a clever idea. It tackles a prevalent problem of diffusion based restoration models - how you provide conditioning without losing important detail or introducing artifacts.

The perceptual quality of results shown is quite impressive. Table 1 shows clear improvement on metrics like MUSIQ and CLIP-IQA. The qualitative results reflect this very well too, by achieving a tough balance between sharpness and fidelity.

The proposed large scale dataset (if released) is also an important contribution to the future development in the deblurring community. Although it is not completely new (built upon existing datasets such as DVD, RSBlur etc.), its careful curation seems effective, as exemplified in the qualitative results.

The details and insights (specially on effect of diffusion model ablations and datasets) provided in the supplementary material are quite informative!



Weaknesses:

Contributions are lacking in terms of novel architectural components or a more profound theoretical insight. While the overall framework is novel, the individual components are not fundamentally so. The paper presents a clever engineering solution by employing relatively known ideas (such as Deep kernel prediction and Element-wise Adaptive Convolution) in a diffusion based formulation. Pixel-adaptive Filter estimation and element wise convolution (in feature domain) for motion deblurring has been proposed and successfully used in a prior research eg. “Spatio-Temporal Filter Adaptive Network for Video Deblurring (ICCV 2019)”. Similarly, joint sharp image and blur kernel estimation (in pixel domain) through diffusion has been proposed in an existing paper: “Parallel Diffusion Models of Operator and Image for Blind Inverse Problems (CVPR 2023)”. Such literature should be included in related works and experiments/analysis.


The paper is primarily empirical and there is a lack of theoretical analysis of why the proposed iterative refinement process should converge or why it is superior to other potential approaches.


The filtered blurred images (obtained by applying the estimated filters on the blurred image latent and passing it to the decoder to obtain an image) are not shown in the paper. Since learning accurate deblurring filters is at the core of the paper, it is important to demonstrate that through visual examples.


The proposed approach’s parameter count and FLOPs (which are crucial for practical applications) have not been discussed or compared with state-of-the-art deblurring methods. Since the proposed method involves two diffusion processes with multiple sampling steps, it would be interesting to compare its overall speed with the best existing methods.


The qualitative results in Figs. do not contain results of existing methods retrained on new dataset. It is difficult to disentangle the effect of model design and new dataset. It would be helpful to know which of these changes contribute more to the significant qualitative improvements over the existing methods.


The proposed method’s key metric scores (MUSIQ, CLIP-IQA) are inferior to the existing approach PASD on real images (shown in Table). Does this indicate overfitting of the model on training data?

The main paper does not provide enough details and experiments on the LKPN architecture. The architecture presented in Fig. 3 only shows a couple of layers while the actual LKPN (as mentioned in the supplementary material) is a UNet with 8 encoder and decoder blocks containing a ResBlock and a self-attention layer. For reproducibility and a deeper understanding of the method, it would be beneficial to know and architectural ablations, and know more details such as specific layers, number of channels, parameters, etc.


Since paper implementation is not provided, extensive qualitative results on key benchmarks should be provided for better assessment and future comparisons.

---

> ### Author Rebuttal · Authors · 2025-07-30
>
> **1. The novelty of this paper**
>
> While it is true that some underlying techniques, such as kernel prediction and adaptive convolution, have been explored in prior research, our contribution lies in the novel integration and application of these concepts within a latent diffusion framework for image restoration, leading to a distinct and effective solution.
>
> Specifically:
>
> 1. Diffusion-based Formulation: The "Spatio-Temporal Filter Adaptive Network" (ICCV 2019) directly estimates kernels from the blurred frame and its adjacent frames in a video sequence, leveraging temporal information. In contrast, our method is specifically designed for the latent space of pre-trained diffusion models. It does not rely on temporal context but instead estimates the degradation kernel by jointly leveraging the input latent of the blurred image and the progressively refined, denoised latent features generated by the pre-trained diffusion model during the iterative process. This integration of the diffusion model's strong generative prior fundamentally differentiates our approach from non-diffusion based methods.
>
> 2. Spatially-Variant Degradation Modeling: Unlike "Parallel Diffusion Models" (CVPR 2023), which estimates a single, global blur kernel, our method focuses on estimating a spatially-variant degradation. This is crucial for handling real-world blurs that vary across the image. Our LKPN network predicts location-specific kernels, enabling more accurate and localized restoration.
>
> 3. Iterative Refinement: We introduce an iterative mechanism where the denoised output from the diffusion model at each step provides a clean prior that actively guides the refinement of the spatially-variant degradation estimate. Non-diffusion methods (as mentioned above) perform "static, one-time" degradation estimation and restoration, whereas our method enables "dynamic, iterative" estimation and restoration. It fully leverages the diffusion model's strength in generating high-quality images. This is further validated by our ablation study in Table 2: the "w/o SD for LKPN" variant, which removes the iterative refinement by not using the diffusion model's features, results in significantly worse performance, confirming the critical role of our iterative design.
>
> We agree that the related work section can be strengthened, and we will include the mentioned papers in the revised manuscript to provide a more comprehensive context.
>
> **2. Why the proposed iterative refinement process should converge**
>
> The process is designed to leverage the inherent property of diffusion models: the restoration quality improves progressively over the denoising steps. As the diffusion model generates increasingly cleaner latent features with each step, these high-quality features serve as a strong prior for our LKPN network. This enables LKPN to estimate more accurate deblurring guidance at each subsequent step.
>
> As detailed in Section 5 of the Supp, we show that our LKPN, combined with the EAC module, generates increasingly accurate deblurring guidance over the diffusion steps, which in turn facilitates better image restoration. This dynamic, iterative interaction is a key advantage of our approach. Figure 10(c) shows that the MANIQA scores of the intermediate results progressively improve from left to right, with values of 0.43, 0.47, 0.55, and 0.58, respectively, clearly demonstrating the quality enhancement throughout the iterative process.
>
> **3. Why DeblurDiff is superior to other potential approaches**
>
> As discussed in L40–L50 and L121-L129 of the main paper, methods like DiffBIR rely heavily on the initial pre-deblurred result. When this initial estimate is poor (Figure 1(c)), it can lead to significant structural errors in the final output (Figure 1(g)).
>
> In contrast, our method features an iterative refinement process, where the LKPN, guided by the denoised output of the diffusion model, progressively generates more accurate degradation estimates, leading to a clearer conditioning input and a feedback loop that produces a more distinct and artifact-free image. (Figure 1(h)).
>
> **4. Visualization of the filtered blurred images**
>
> In Figure 2 of the main paper, we show the intermediate results of the reconstructed latent clean images at different diffusion steps. These results are obtained by applying our estimated deblurring kernels in the latent space and then decoding them, effectively serving as visualizations of the filtered blurred images. Furthermore, Figure 10 in the Supp provides additional ablation studies that include these intermediate reconstructions.
>
> We will provide a more detailed discussion and clarification on these visual results in the revised paper.
>
> **5. Comparison of Params, FLOPs, and runtime.**
>
> We have provided runtime information for some methods in Table 4 of the Supp. We provide comprehensive comparisons of model parameters and FLOPs in the following table.
>
> | |FFTformer|ResShift|DiffBIR|PASD|Ours|
> |-|-|-|-|-|-|
> |Params (M)|16.6|119|1717|1900|1845|
> |FLOPs (G)|131|5491|24234|29125|25526|
> |Times (s)|0.8|15|55|52|46|
>
> As discussed in L86–L92 of the Supp, FFTformer is not a diffusion-based method, which makes it faster but results in inferior visual quality. ReShift does not use a pre-trained diffusion model, giving it some advantages in runtime over methods that do; however, its visual performance is worse than diffusion-based methods such as DiffBIR, PASD, and DeblurDiff. Among all methods that leverage pre-trained diffusion models, our approach achieves both higher fidelity (as shown by better PSNR and SSIM in Table 1) and better perceptual quality. Moreover, our method has fewer parameters and lower FLOPs than PASD, and runs faster than both DiffBIR and PASD.
>
> **6.  The qualitative comparisons lack baseline methods retrained on the new dataset**
>
> The qualitative results (e.g., in Figures 1-2 and Figures 4-6) are consistent with Table 1, where  DiffBIR and PASD were retrained on our dataset for fair comparison. In the revised manuscript, we will add an asterisk (*) to DiffBIR and PASD in the figure labels to clarify this.
>
> **7. Overfitting of the model on training data**
>
> In fact, as stated in line 217 of our paper, our training set includes only RSBlur and MCBlur, while the test sets consist of five distinct datasets: GoPro, DVD, RealBlur, RWBI, and Real Images, ensuring a strict train/test split and avoiding data leakage. Furthermore, both RWBI and Real Images are real-world blur benchmarks without ground truth, making overfitting to GT impossible. Notably, on the RWBI dataset, our method achieves the highest scores in MUSIQ, MANIQA, and CLIP-IQA among all compared methods. This demonstrates the strong generalization capability of our model rather than overfitting.
>
> **8. More details and experiments on the LKPN architecture.**
>
> In the revised manuscript, we will provide a comprehensive description of the LKPN architecture. Specifically, LKPN is a U-Net with 4 stages (often referred to as 4 levels), resulting in a total of 8 encoder and decoder blocks (2 blocks per stage). Each encoder block consists of a ResBlock followed by a self-attention layer, and each decoder block consists of a ResBlock followed by a nearest-neighbor upsampling layer. After each encoder block, a downsampling layer is applied to reduce the spatial resolution before feeding the features to the next stage.
>
> The input channel is 8, and the number of hidden channels is 128 throughout the network. The total number of parameters is 144M.
>
> Due to time constraints, we were only able to conduct ablation studies on different parameter scales of the LKPN network. The results are shown in the table below, where a larger LKPN with more parameters brings moderate performance improvement, indicating a positive correlation between model capacity and restoration quality.
>
> ||NIQE|MUSIQ|MANIQA|CLIPIQA|
> |-|-|-|-|-|
> | LKPN_C64|3.9463|51.7462|0.5861|0.5416|
> | LKPN_C128|3.6628|52.9263|0.5963|0.5496|
>
> We commit to releasing the full code and model configuration files to ensure complete reproducibility.
>
> **9. Detailed implementation of the proposed method**
>
> We are actively preparing the code, model weights, and corresponding test results. We commit to releasing the code to ensure reproducibility and facilitate future research.
>
> **10. Replacing convolution operations with attention**
>
> Our goal is to estimate a spatially variant kernel to guide the restoration of sharp images in the latent space, effectively preserving the information from the blurry input. We have explored replacing this dynamic filtering operation with a cross-attention mechanism. However, as shown in the table below, this alternative design resulted in inferior performance. Therefore, we find that the dynamic filtering approach is more effective for this task. This remains an interesting direction for future exploration.
>
> | |NIQE|MUSIQ|MANIQA|CLIPIQA|
> |-|-|-|-|-|
> |Cross Attention|4.2351|50.3223|0.5642|0.5257|
> |EAC|3.6628|52.9263|0.5963|0.5496|
>
> **11. Adding a pre-deblurred image as extra input**
>
> We have experimented with this approach by retraining our model to accept additional latents extracted from a pre-deblurred image (generated by a pre-trained DRM) as input, concatenated with the original conditioning features. The experimental results are shown in the table below.
>
> | |NIQE|MUSIQ|MANIQA|CLIPIQA|
> |-|-|-|-|-|
> | DeblurDiff|3.6628|52.9263|0.5963|0.5496|
> | DeblurDiff w/ DRM|3.5472|53.8815|0.5852|0.5581|
>
> Adding the DRM input leads to improved performance in NIQE, MUSIQ, and CLIP-IQA, while MANIQA slightly decreases. This indicates that incorporating the pre-deblurred image as an additional conditioning signal can enhance the model's perceptual quality to some extent.
>
> **12. Whether or not the FFTformer used in DiffBIR is retrained on the proposed dataset.**
>
> Yes, the FFTformer used in DiffBIR was retrained on our proposed dataset. We will clarify this point in the revised manuscript.

---

> > ### Author Response · Authors · 2025-08-06
> >
> > Dear Reviewer,
> >
> > As the discussion period is coming to a close, please let us know if you have any additional questions or comments regarding our previous rebuttal. We would be happy to provide further clarification or explanations if needed.
> >
> > Best,
> >
> > Authors

---

> > > ### Comment · Reviewer_qdjB · 2025-08-09
> > >
> > > Thank you for the detailed rebuttal. It does address several of my concerns.
> > >
> > > One of the key concerns still remains: about the lacking fundamental novelty in technical contributions. However, the generalized design and effectiveness of the overall framework is praiseworthy. I continue to lean towards acceptance.

---

### Comment · Area_Chair_s8S9 · 2025-08-03
**Author-Reviewer Discussions**

Dear Reviewers and Authors,

Thank you all for your efforts. The author-reviewer discussion phase is open until **August 6, 11:59 PM AoE**.

This paper has received mixed ratings, and the authors have provided detailed rebuttals.

Reviewers, if you haven’t yet, please kindly take a moment to check the response and let us know if your concerns have been addressed.

Thanks,

The AC

---

### Decision · Program_Chairs · 2025-09-17

**Decision:**

Accept (poster)

**Comment:**

This paper introduces a latent kernel prediction strategy within Stable Diffusion for image deblurring. The submission received mixed ratings (3×Accept and 1×Weak Reject). After the rebuttal stage, most of the concerns raised by Reviewers qdjB, 34aF, and TarL have been largely addressed. However, Reviewer Xgdm maintained strong concerns regarding the evaluation validity and the incremental performance.

After carefully reading the paper, reviews, and all discussions, the AC shares concerns about the originality of the key components and the performance, as also highlighted by Reviewers qdjB and Xgdm:

- On EAC originality (qdjB): The proposed Element-wise Adaptive Convolution (EAC) layer appears nearly identical to the Filter Adaptive Convolutional (FAC) layer from *“Spatio-Temporal Filter Adaptive Network for Video Deblurring (ICCV 2019)”*. Upon examination, the AC observed that the EAC figure (Fig. 3) is essentially the same as FAC (Fig. 3 in that paper), and the equation and description in supplementary (Lines 49–53) almost replicate FAC’s formulation (Eq. 1 and its explanation). Yet, the submission does not cite or discuss this prior work. Such an omission raises concerns and may mislead reviewers into overestimating the novelty of the contribution, and therefore EAC cannot be considered a new component of this paper. Moreover, the ablation study in the authors’ response (Q4 by Xgdm) indicates that performance heavily depends on EAC, removing it (“w/o EAC”) reduces PSNR from 24.32 to 22.57. This further raises concerns about the true contribution and originality of the method.

- In addition, Reviewer Xgdm questioned that degradation estimation has already been widely studied in image restoration. The authors argue their distinction lies in “iterative refinement.” The AC agrees this is an interesting and reasonable design to introduce into the diffusion framework, though it is not a fundamentally new concept and has been explored in earlier works such as *“Blind Super-Resolution With Iterative Kernel Correction (CVPR 2019)”*.

- On performance and efficiency (Xgdm): The most serious concern raised by Reviewer Xgdm is that the performance improvements are incremental and do not demonstrate clear progress over the state of the art. The AC also share this concern, competing methods such as Hi-Diff and ResShift surpass the proposed approach on multiple metrics (Table 1), while being significantly more efficient in terms of parameters, FLOPs, and inference time. For instance, according to the authors’ response, Hi-Diff is ~50× faster with only 1/200 FLOPs, yet still outperforms the proposed method on several benchmarks.

In summary, this paper received three positive ratings and introduces an interesting integration of iterative kernel refinement into a diffusion framework for image deblurring. At the same time, the AC notes that the novelty may be overestimated due to the omission of prior work, particularly with the key component EAC showing strong similarity to the earlier FAC design but without proper discussion or acknowledgment. While the quantitative improvements over recent state-of-the-art methods are not always substantial, the approach provides clear gains in visual quality, which is an important strength for practical deblurring applications. For the final version, the AC encourages the authors to explicitly cite and clarify the relation to prior works (e.g., FAC for EAC) and to more clearly position their contributions.